# Graphene Oxide-Based Nanofiltration for Hg Removal from Wastewater: A Mini Review

**DOI:** 10.3390/membranes11040269

**Published:** 2021-04-08

**Authors:** Megawati Zunita

**Affiliations:** Department of Chemical Engineering, Faculty of Industrial Technology, Institut Teknologi Bandung, Jalan Ganesha 10, Bandung 40132, Indonesia; m.zunita@che.itb.ac.id

**Keywords:** mercury, nanofiltration, graphene oxide

## Abstract

Mercury (Hg) is one of heavy metals with the highest toxicity and negative impact on the biological functions of living organisms. Therefore, many studies are devoted to solving the problem of Hg separation from wastewater. Membrane-based separation techniques have become more preferable in wastewater treatment area due to their ease of operation, mild conditions and also more resistant to toxic pollutants. This technique is also flexible and has a wide range of possibilities to be integrated with other techniques. Graphene oxide (GO) and derivatives are materials which have a nanostructure can be used as a thin and flexible membrane sheet with high chemical stability and high mechanical strength. In addition, GO-based membrane was used as a barrier for Hg vapor due to its nano-channels and nanopores. The nano-channels of GO membranes were also used to provide ion mobility and molecule filtration properties. Nowadays, this technology especially nanofiltration for Hg removal is massively explored. The aim of the review paper is to investigate Hg removal using functionalized graphene oxide nanofiltration. The main focus is the effectiveness of the Hg separation process.

## 1. Introduction

Mercury (Hg) is one of heavy metals with the highest toxicity and negative impact on the biological functions of living organisms (mainly humans) [1]. Mercury contamination in the environment pollutes water systems mainly due to atmospheric deposition (e.g., rainfall) and effluents from industrial processes primarily as Hg [2,3]. There are a lot of potential industries that contribute to mercury pollution such as chloroalkyl compound, vinyl chloride, plastics, electrical equipment, batteries, pulp and paper, and paint manufacturing [4]. 

Hg pollution in waterways is a well-known problem and some countries such as U.S.A., Brazil, Indonesia, India, Iraq and China have detected mercury at harmful levels [5,6,7]. Potentially harmful concentrations of Hg has also been observed in drinking water supplies [6,7] as well as reservoirs that could serve as drinking water sources [8,9,10]. The World Health Organization has set 1 ppb as the maximum concentration of mercury in drinking water while the US EPA has set 2 ppb. The challenges associated with mercury removal are the generation of brine solutions or waste products that have to be disposed of or need regeneration, adding an additional process [11]. Therefore, Hg recovery in a more concentrated form is preferred and offers more benefits. Hg is typically present in very low concentrations along with other pollutants that will compete in e.g., chemical precipitation, ion exchange or adsorption processes, increasing the amount of material that must be processed to remove Hg. This introduces secondary pollution when chemical or biological treatment is used, lowering drinking water quality

Membrane technology has experienced rapid development of late, mainly due to its advantages and potential for various applications in many sectors [12,13,14,15,16,17,18,19]. In addition to providing a selective layer for one of the reaction components, the membrane can also act as a catalyst support and even be catalytically actively itself. Therefore, membrane technology is widely applied in the world of synthesis and waste treatment [20,21,22,23,24,25].

One of the membrane technology techniques considered to separate ions is nanostructured membrane technology. Graphene is one of the materials that has a nanostructure can be used as a thin and flexible membrane sheet with high chemical stability and high mechanical strength [23,24,25,26,27,28,29,30]. Monolayer graphene membranes are accepted to be able to remove metal ions very effectively and efficient for wastewater due to their nanopores [31,32]. Graphene membranes with functionalized nanopores have proven separation performance. The metal ion separation performance is promoted by nanopore size, temperature, driving pressure and carboxyl groups on the membrane surface, which increase the selectivity [31,32]. In addition, one of the graphene derivatives that is most applied to separate metal ions is graphene oxide (GO), which has a two-dimensional structure. Nowadays, GO membranes are a widely used kind of nanomaterial sheet for wastewater processing in industry due to their high selectivity properties for the separation the matrix ions of samples. Besides, many studies have modified the surface of the GO membranes, therefore their metal ion separation performance is increased significantly [33,34]. Furthermore, the laminated GO functions as a two-dimensional water channel due to its planar construction, good dispersity, and hydrophilicity [35,36]. GO membranes have nano-channels to provide ion mobility and molecule filtration properties [37].

Mercury removal has been long term task for industries. There are various ways to remove mercury from wastewater so that it will not end up in our drinking water. This paper will briefly discuss general trends in mercury removal from aqueous solutions. Special attention will be given to nanofiltration using GO membrane materials. The effectivity of separation and also the benefits and drawbacks of GO-based nanofiltration will also be deeply discussed in the next sections.

## 2. The Toxicity of Mercury and Its Removal

Mercury is a neurotoxin that can cause damage to the central nervous system. High concentrations of mercury cause impairment of pulmonary and kidney function, chest pain and dyspnoea [38]. The classic example of mercury poisoning is the Minamata Bay incident in Japan [39]. Moreover, Hg accumulated in the body of organisms can attack the central nervous system, and excess exposure of the body for a long time can have a hard impact on human organs such as brain damage, gastrointestinal damage, and in extreme cases, death [11]. The dangers of mercury exposure have led to an increase of international restrictions on mercury levels in waterways [11]. Mercury can be present as elemental mercury (Hg^0^), oxidized mercury (Hg^2+^), and particulate mercury (Hgp) [40,41]. All orbitals of Hg^0^ are filled with electrons and it has no unoccupied orbitals (5d106s2 outer electron configuration) [42]. This makes Hg^0^ the most difficult species to remove due to its very high volatility, low water solubility, and relatively inertness [43,44,45,46]. On the other hand, Hg^2+^ is water soluble and can easily enter water bodies and be converted into methyl mercury (MeHg) and then accumulate in living organisms, including humans [47,48]. Hgp has a relatively short atmospheric lifetime and usually spreads along with flue gasses [49], causing respiratory and chromosome damage [45,50].

Mitigation of mercury pollution of drinking water can be done by direct treatment of drinking water or treatment of pollution sources such as industrial wastewater streams where Hg concentrations are much higher. The common methods to remove mercury from wastewater are precipitation [51], cementation [4], ion exchange [52], adsorption [53], nanofiltration [54], and solvent extraction [55]. Slow kinetics, low capacities due to heterogeneous reactions and interface transfer are the main limitations of said methods that make development of new techniques for mercury separation interesting [56,57]. In addition, adsorption of mercury can be carried out using several materials. Among sorbent materials, activated carbon is a commonly used sorbent because of its high removal capacity [58]. Moreover, addition of chlorine-, iodine- and sulfur-treated activated carbon boost its capacity to capture elemental mercury [59], and the efficiency of mercury removal can also be enhanced by increasing the oxygen concentration [60]. Zhang et al. [54] reported that a sulfur-functionalized polyamide-based nanofiltration membrane can effectively reduce Hg^2+^ concentrations in drinking water sample from 10 ppm to a low level of 0.18 ppb where the acceptable limit of Hg in drinking water is around 2 ppb. Single metals can also be used to adsorb mercury. Copper and some noble metals such as gold, silver, platinum and palladium have also been used for mercury removal. In particular, gold is preferred for Hg removal due to its efficiency. Moreover, gold is more immune to impurities such as organic substances or sulfur-containing species [61]. On the other hand, the efficiency of Hg removal is highly dependent on the temperature [62]. Metal oxides such as Fe_2_O_3_, CuO and CaO also exhibit significant mercury removal ability [63]. Specifically, MnO_2_-based materials have high efficiency for mercury removal, better regeneration, and high activity for a long time [64]. Other materials such as surfactants containing oxygen, nitrogen, and phosphorous also show promising mercury removal capacity [65].

Another method to remove mercury is selective catalytic reduction (SCR). By SCR, Hg^0^ is oxidized into Hg^2+^ that is easier to remove. Moreover, the catalyst will bind chemically with mercury so that the water effluent will be mercury-free. Gold is a promising catalyst for SCR by chlorine. Cl_2_ can easily chemisorb on the Au surface and will easily oxidize mercury [66]. V_2_O_5_ is another important SCR catalyst. The presence of HCl, strongly influence mercury adsorption and oxidation on vanadium catalysts [65]. H-ZSM-5-supported Fe and Cu have been synthesized for SCR of mercury as Fe/HZSM-5 and Cu/HZSM-5 have strong ability for Hg^0^ removal [67,68].

## 3. Membrane Separation for Mercury and Heavy Metals

Many researchers have developed various solutions in the area of mercury separation from wastewater [69,70]. Several methods are used to remove mercury such as adsorption, extraction, electrolysis, and modern ones with better performance, e.g., membrane technology [11,70,71,72]. Membrane technology (ultra-, micro- and nanofiltration) work based on the selectivity and the pore size of membranes. The separation mechanism firstly involves adsorption prior to extraction or rejection of chemicals from the permeate part [73,74,75,76]. Adsorption is frequently used for metal ion separation due to its low cost, simple design, and mild operation conditions.

In a membrane-based separation process, the membrane itself acts as a contactor layer through which ion complexes and particles pass via diffusion [73,77]. The nanopores of graphene membranes provide a significant pathway for ion penetration, therefore the ion selectivity facilitates the metal ion separation, Furthermore, the ion diffusion of porous graphene membranes can be enhanced by acid addition. To develop a metal ion separation performance, the pores of a graphene membrane can be modified by using oxide functional group derivatives [78,79,80]. The membrane modules often used in separation of mercury and heavy metals are hollow fibers and sheet layers [71,81,82]. Previous researchers have investigated various thickness and pore sizes of the membrane to enhance the mobilization of metal ions while using an immobilized solvent to achieve a more selective separation process [72,83,84]. However, a thick membrane still has advantages i.e., in the form of a transverse flow contactor which is better than the parallel flow contactor that could be more unstable. Besides, a supported liquid membrane method has drawn attention as an alternative in extractive separation for metal ion removal or neutral molecules from dilute solution. Such a method is more simple and offers advantages compared to conventional extraction methods [75,85,86]. Moreover, emulsion liquid membranes are also often used to separate mercury from wastewater and metal ion mixtures supported by trioctylamine (TOA) [74,87,88] and bulk liquid membranes represent a potential method [83]. Furthermore, ion exchange membranes are another potential method for mercury removal [89]. 

A lot of research has been done in the field of mercury removal and given good membrane utilization performance. Several materials can be used to separate of mercury and heavy metal ions via membrane filtration (e.g., using polyaniline, polypyrrole, cellulose triacetate, nylon, chitosan, polypyrrole, polyethersulfone, graphene-based membranes, zeolite-based membranes, polyvinylamine, etc [69,73,77,81,90,91]). Those kinds of polymers have the properties which are required in membrane-based separation for mercury removal such as low cost, high stability, high selectivity for mercury ions, thermal stability, high chemical resistance, good ion-exchange capacity, reproducibility, high selectivity for heavy metals, the possibility of forming coordinating ligands with mercury and adsorb anions through electrostatic interaction or hydrogen bonding [85,92,93,94,95]. 

To get good separation results via a membrane-based process the separation process is designed based on the sample source of mercury and the characteristics of the sample as main factors. As an example, Koopman et al. [96] used a hollow fiber membrane contactor for heavy metal separation in the phosphoric acid industry where its sample preparation and conditions were optimized. The examples of treatments for mercury come from the chlor-alkali industry, electrical and electronic industries (in the manufacture of mercury vapor lamps and fluorescent tubes, batteries, electric switchgears, etc.), plastics industry (in the manufacture of vinyl chloride), paper and pulp industry and pharmaceutical industries which each have suitable conditions for mercury separation. 

Some authors have already reported membrane-based separation processes for mercury and heavy metals in various industries. Khan et al. [91] have reported a polypyrrole polyantimonic membrane with acid-based ion exchange which is highly selective for mercury ion extraction. Some important divalent ions including Hg^2+^–Zn^2+^, Hg^2+^–Ni^2+^, Hg^2+^–Cu^2+^, Hg^2+^–Fe^3+^, Hg^2+^–Cd^2+^, Hg^2+^–Mg^2+^, etc. were separated using an organic-inorganic composite system [91]. In another study [97] mercury was separated using a supported catalytic membrane, e.g., a Mn/Mo/Ru/Al_2_O_3_ membrane which achieved 95% Hg removal at 423 K. Moreover, Ura et al. [74] used Nylon 6,6 as a support, trioctylamine (TOA) as a carrier and dichloroethane as the solvent to separate mercury and lignosulfonate. The results showed that the removal of mercury and LS from mixtures was about 52.6% and 50.2%, and even in pure solution an 81% removal was achieved. Huang and co-workers [98] separated mercury using polyvinylamine as the mercury-binding polymer which achieved 99% removal. In their study, the ultrafiltration technique was used. The separation occurred on the surface of the amine polymer that created binding between mercury and the polymer. A graphene-based membrane that used graphene as nanostructured membrane with good mercury removal performance was reported by Jafar and co-workers [81]. A summary of the performance of different membrane separation techniques for mercury and heavy metals is shown in Table 1 and further discussion of graphene-based membrane (nanofiltration) will be provided in the next section.

## 4. Graphene-Based Membranes

Graphene is a novel material that consists of a one layer honeycomb-like carbon structure (Figure 1). Thus, it is known as an ultrathin two-dimensional material regarding its one-molecule-thickness. Consequently, graphene has very unique properties as an ultra-thin, light, transparent [104,105] yet mechanically strong and thermally stable material [106,107]. Moreover, graphene is also reported to have a good optical [108] and electrical [106,109] properties. Many researchers have functionalized graphene in order to improve its performance by introducing graphene-based derivatives, including graphene oxide (GO), reduced graphene oxide (rGO) and other composites. In composite forms, graphene may be strengthened by addition of other materials [110,111,112,113,114] or strengthening a conventional material by incorporation of graphene [115].

Edwards and Coleman [116] categorized two types of graphene synthesis, namely bottom-up and top-down. In a bottom-up process, graphene is formed via reformation of some other component (mostly silicon carbide). On the contrary, in a top-down process graphite is exfoliated into a single layer graphene. Most researchers refer to the infamous Hummers [117] method as the top-down graphene synthesis, especially for graphene oxide. The improvement of Hummers-based method has been of interest for some researchers [2,118,119] to make it more feasible for massive production. More specifically, the synthesis method may affect the properties of the resulting graphene. Therefore the modifications of synthesis methods should consider the intended application of the graphene itself regarding its required properties [106].

Due to its unique properties, graphene and its derivatives have been explored in a wide range of applications, including electrical devices [104,109], adsorbents [112,120,121] and also as a separation membrane [122]. Considering its very small openings between the carbons, a perfect graphene sheet is impermeable to a lot of gases as small as helium [122]. However, researchers have modified graphene by creating holes to make it semipermeable to certain gases. The porous graphene sheet has been developed and reported to have a very high selectivity for hydrogen in the presence of many other gases including methane, nitrogen, carbon dioxide, oxygen, ammonia, and argon [105,123,124,125,126].

In liquid applications, graphene-based membranes are mostly used in multilayer form [127,128,129,130]. The transport mechanism across the membrane utilizes the imperfections of the graphene sheet to create a channel for water (or another solvent). The defects include holes, wrinkles, inter-edge and inter-layer spaces [122,127,130,131,132,133]. The modification of graphene membranes in this application is related to widened water channels within the multi-layer membrane [122,131] which thus increase the flux while still considering the affected separation properties (e.g., rejection, selectivity). The modifications that have been reported includes the utilization of GO [131,133,134,135,136,137], creating holes within the sheet [122], increasing the space between layers, which can be done via crosslinking or incorporating carbon nanotubes [120,138] and synthesizing the composites [139]. Compared to pure graphene, GO is reported to have more functional groups on the surface [111,131] hence widening the interlayer channel and increasing the flux. SEM photos (Figure 2a–d) show that the GO composite membrane is more dense due to layer by layer interactions. A schematic of a GO membrane for wastewater treatment is presented in Figure 2e.

Graphene membranes are reported to work within the nanofiltration range which is suitable for separation of ionic species [127,134,137,140], metals [141,142], and also dyes [133,136,137,138,143]. In some references it was shown that nanofiltration process (the exclusion mechanism) depends on the steric, electric and dielectric properties of the metal ions [127,134,137,140]. Similar to most nanofiltration membranes, the separation process in a graphene nanofiltration membrane occurs by two types of mechanism, namely sieving and Donnan exclusion. In the sieving mechanism, the comparison of molecule size and the pore size does matter. The membrane will totally reject molecules which are bigger than the membrane pore size. In the Donnan exclusion mechanism, the separation considers the interactions between the membrane and the solutes related to their charges [144]. Unlike the original graphene, a GO membrane is charged, thus giving better performance in rejecting ionic species, including dyes [137]. 

## 5. Graphene Oxide-Based Nanofiltration Membrane Preparation

In general, the main route for making GO is chemical oxidation and exfoliation of graphite using the Brodie, Staudenmaier, or Hummers methods. Brodie reported that an oxidizing KClO_4_ solution with fuming HNO_3_ can form GO only with graphite carbon containing a graphite-structured region. Staudenmaier showed that GO formation occured when graphite was reacted with strong acids (i.e., H_2_SO_4_, HNO_3_, and KClO_4_). Hummers and Offeman found a very practical method to prepare GO using H_2_SO_4_ and KMnO_4_ [35,145]. At present, GO is usually synthesized according to the modified Hummer method in which the rate of reaction is carefully controlled to keep the reaction temperature below 20 °C. The appearance of GO and graphene can be analyzed by SEM, as seen in the example results shown in Figure 3. The GO profile is more rigid and thick than that of graphene, therefore GO is very promising as a high selectivity membrane to separate Hg from wastewater.

Several review papers have described the preparation of Fe_3_O_4_/GO nanocomposites by two different routes: impregnation (denoted as mGOi) and coprecipitation (denoted as mGOp). In particular, Fe_3_O_4_ nanoparticles can be synthesized by the Massart method with mixtures of FeCl_3_·6H_2_O and FeCl_2_·4H_2_O heated to 60 °C. The clear yellow solution product is separated under vigorous agitation. Then, aqueous ammonia solution is added to the solution until the pH of the solution reaches 10. The reaction was maintained for an additional 30 min under vigorous stirring. Nitrogen was used as the protective gas throughout the experiment. After completion of the reaction, the resulting black precipitate was collected by an external magnetic field, followed by washing several times with water and ethanol. Finally, the Fe_3_O_4_ nanoparticles were freeze-dried. The next step is GO membrane preparation by using a coating method. A GO aqueous solution was made by dissolving GO powder in deionized water. To form the GO nanosheets, the GO solution was subjected to ultrasonic irradiation several times. Usually, a ceramic hollow fiber membrane is used as GO membrane material due to its properties like being easily stacked on the surface with pressure driving. Then the as-prepared GO membrane was dried in a vacuum chamber at 40 °C and is ready to be used for Hg separation [28,127,146].

## 6. Utilization of Graphene Oxide-Based Nanofiltration for Mercury Removal

The method of mercury separation via membrane filtration has been discussed in the previous section. Besides the mentioned sieving and Donnan exclusion mechanisms, the separation process in a membrane system can be enhanced by introducing external forces or modifying the component of interest. Some researchers introduce other agents such as polyethyleneimine (PEI) [147] or iron sulfide [93] to form a complex thus promoting the separation process. These complexes may either be bigger in size or have a special interaction with the membrane surface hence promoting the separation. A complex with bigger size allows the process to be performed in the ultrafiltration range thus the utility requirements are less.

Besides modifying the aqueous mercury into a complex species which has special surface properties, it is also possible to modify the membrane thus enhancing the separation process. Some researchers [81,148,149,150] have introduced functionalized graphene sheets constructed by holes in a graphene sheet which is modified by some agents that improve the surface properties. Figure 4 illustrates a functionalized graphene where the holes are modified by other functional groups. The functional groups introduced in the graphene holes include chlorine [81], xanthate [149] and thiol groups [150]. Besides the holes, the surface of the graphene membrane itself can be modified forming a composites, such as an iron-graphene composite [149]. 

Even though functionalized graphene sheets offer special affinity for aqueous mercury (or its complexes), the functional groups themselves may form a barrier that hinders the passage of the mercury. Azamat [81] reported that an electrical force was required in order to support the mercury transfer across the membrane. In another case, Cui [149] used a magnetic force to enhance the separation. Besides those external factors, some process parameters such as pH and ion concentration also affect the separation performance. The performance of graphene oxide-based membranes for Hg removal is summarized in Table 2.

Utilization of membrane materials in order to modify the interaction properties between the solute and membrane surface is closely related to adsorption processes. In other cases, a graphene-based material showed good affinity for adsorption processes for organic dyes [112,145,151]. Furthermore, similar materials also give good performance in nanofiltration processes [120,127,131,138]. Studies of mercury adsorption using graphene and its derivatives have been carried out by some researchers. Modifications have also been performed thus an effective removal higher than 99% was achieved [149,150,152,153,154]. 

Recently, research in mercury removal via graphene oxide nanofiltration has been limited to functionalized graphene sheets [81,149]. However, there is huge potential to utilize graphene oxide-based membranes in other configuration which are discussed in the previous section. Other studies of adsorption techniques have already presented good results. Referring to graphene oxide membrane applications in organic dye removal, many studies were conducted following good results in adsorption processes. Thus, mercury separation using graphene oxide-based membranes seems to have great potential.

## 7. Conclusions

Hg is one of the highest toxic substances that should be removed from any wastewater prior to environmental discharge. Several techniques can be used to perform the removal, including adsorption, liquid extraction, electrolysis, and membrane separation processes (i.e., more efficient and effective techniques). The membrane separation process can be improved by adding complexes or selecting a proper membrane material, including oxide derivatives. Graphene oxide-based membrane has presented excellent performances in nanofiltration processes for Hg removal. Recent utilization of graphene oxide-based membranes in Hg separation is only limited to functionalized graphene sheets, therefore it needs widely more improvement. Several graphene membrane types can be developed and have big potential for Hg removal such as layered membranes, intercalated membranes, crosslinked, and also composites.

## Figures and Tables

**Figure 1 membranes-11-00269-f001:**
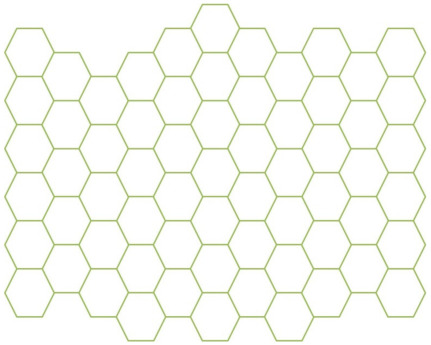
Structural Illustration of a Graphene Sheet.

**Figure 2 membranes-11-00269-f002:**
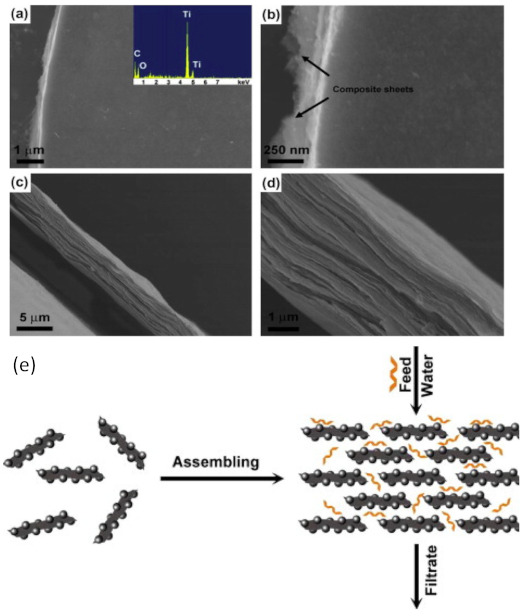
(**a**–**d**) SEM photos of a GO composite membrane and (**e**) a schematic of a GO composite membrane for wastewater treatment. The figures are reproduced with permission from [111]. Copyright Elsevier, 2021.

**Figure 3 membranes-11-00269-f003:**
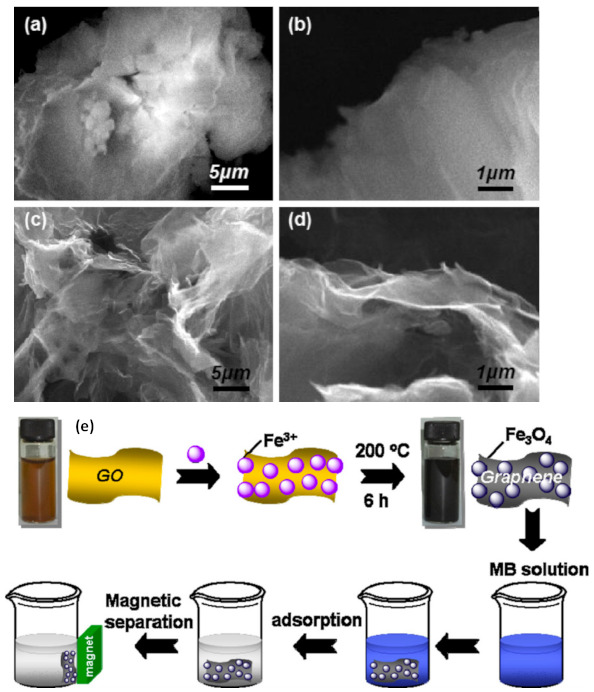
The SEM photos of GO (**a**,**b**) and graphene (**c**,**d**), and a schematic of GO membrane preparation (**e**). The figures are reproduced with permission from [112] Copyright Elsevier, 2021.

**Figure 4 membranes-11-00269-f004:**
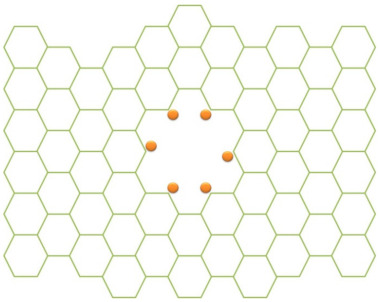
Illustration of a functionalized graphene sheet. This Figure is adopted and reproduced with permission from [81]. Copyright Elsevier, 2021.

**Table 1 membranes-11-00269-t001:** Performance of Membrane Separation for Mercury and Heavy metals.

Membrane	Mixture Components	Separation Conditions	Rejection % or Removal %	Ref.
Two micro-porous PP supported membrane loaded with a mixed N/O/S-donor	Ag^+^ and Hg^2+^	Na_2_S_2_O_3_ (0.04 M) and EDTA disodium salt (0.025 M) as stripping agents in 3.5 h	95.3% Ag^+^ and 94.7% Hg^2+^	[75]
30% cellulose triacetate (CTA), 60% 2-nitrophenyl octyl ether (NPOE), and 10% *w*/*w* Cyanex 471	HgCl_2_	Hg [2+] in HCl + NaCl at pH 12.	81 %	[82]
D2EHPA (CAS No. 298-07-7) with 98.5% purity	HgCl_2_	1 M H_2_SO_4_ with 0.5 M thiourea	92%	[84]
polyethyleneimine (PEI)	Hg in a heavy metal mixture	pH 5.5 cadmium/polymer ratio about 0.35 mercury/polymer ratio about 0.39	98% Hg and 97% Cd	[99]
1,1,7,7-tetraethyl 4(tetradecyl)diethylenetriamine (TE14DT)	Cd^2+^/Pb^2+^ and Hg^2+^/ Cu^2+^ mixtures	pH 2.5	90%	[100]
poly(benzylsulfone)	Hg^2+^	Diluted in hydrochloric acid	>90%	[101]
Mixed-matrix membranes (sorbent particles and polysulfone)	Ca^2+^, Ag^+^, Hg^2+^	Diluted in HCl at pH 4	95% Hg^2+^	[102]
Cyanex 302 (bis(2,4,4-trimethyl- pentyl)thiophosphinic acid) in kerosene	Cu^2+^ and Hg^2+^	Diluted in phosphoric acid slurry	70%	[96]
Mn/Mo/Ru/Al_2_O_3_ membrane	Hg	Diluted in hydrochloric acid	95%	[97]
Polyvinylamine	mercury - sodium chloride and sulfate	feed mercury concentration range tested (0–50 ppm)	99%	[98]
Cross flow membrane filtration cell (CF 042, Sterlitech, California)	Hg^2+^	higher operating pressures (≥34.5 bar)	95%	[103]

**Table 2 membranes-11-00269-t002:** The performances graphene oxide-based membranes for mercury Removal.

GO Membrane Type	Components Mixture	Separation Conditions	Rejection % or Removal % of Mercury	Ref.
Xanthate functionalized magnetic graphene oxide (Fe_3_O_4_-xGO)	Hg^2+^ and methylene blue	pH: 7.5, 3 h, 298 K, 1 atm	94.5%	[149]
Graphene-Diatom (GN-DE) Hydrogel Decorated with αFeOOH Nanoparticles	Hg^2+^	pH: 10, 90 min, 298 K, 1 atm, pore size: 0.22 µm	80%	[150]
mercapto-grafted graphene oxide–magnetic chitosan (GO–MC)	Hg^2+^ in environmental water samples	60 mg of sorbent, pH of 6.5, 10 min for adsorption time, 3 mL of HCl (0.1 mol L^−1^)/thiourea (2% *w*/*v*) as the eluent , 298 K, 1 atm	95% to 100%	[152]
GO foams	Hg^2+^	A small dose of 3DGON (10 mg L^−1^), pH: 5 and 9, 24 h, 298 K, 1 atm	96%	[153]
Three-dimensional reduced-graphene oxide (3-D RGO) hydrogel	Hg^2+^ and F^−^	pH: 6, 24 h, 298 K, 1 atm	65%	[154]

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
