# Peer review of "Graphene Oxide-Based Nanofiltration for Hg Removal from Wastewater: A Mini Review"

_membranes, 2021, doi:10.3390/membranes11040269_

Round 1

Reviewer 1 Report

Title: Graphene Oxide Based Nanofiltration for Hg removal in Wastewater: Mini review

Author: Megawati Zunita

The present work is a short review dealing with Hg removal in wastewater by using nanofiltration membranes (Graphene oxide based).

After introducing the problems and challenges, the author describes the toxicity and the different techniques for Hg removal. In a third paragraph, the author described the membrane techniques used for heavy metals removal (membranes used, treated wastewater, etc.) and gives some interesting results (summarized in table 1) in terms of rejection rates. The two following parts introduce the graphene based membrane and the preparation. The next part of this work focuses on the utilization of this membrane for Hg removal.

This paper is clear, well supported by the literature and the results are interesting for academic and industrial communities

 Accordingly, I recommend the publication of this paper in Membranes after minor revision

Comments:

  • Part 6: this part lacks of information / data in terms of operations, Hg removal (and the rest of minerals?)
  • The introduction part speaks about “drinking water”. Pages 4 and 5 only deal with industrial wastewater treatment. It should be interesting to present one example of drinking water treatment by nanofiltration.

I’m not sure, that in this case, nanofiltration is an appropriate treatment (removal of Hg and heavy metals, but other minerals too).

  • In part 5 or 6, data of graphene oxide based membranes should be interesting for comparison, (hydraulic properties, mean pore radius, electric surface charge, IEP, rejection rate of uncharged solute, etc.)
  • Line 234 to 243: in the literature about nanofiltration, the exclusion mechanisms are steric, electric and dielectric Two were just discussed !

Minor comments:

Line 27, different font sizes.

Author Response

Reviewer #1

Comments and Suggestions for Authors

Title: Graphene Oxide Based Nanofiltration for Hg removal in Wastewater: Mini review

Author: Megawati Zunita

The present work is a short review dealing with Hg removal in wastewater by using nanofiltration membranes (Graphene oxide based).

After introducing the problems and challenges, the author describes the toxicity and the different techniques for Hg removal. In a third paragraph, the author described the membrane techniques used for heavy metals removal (membranes used, treated wastewater, etc.) and gives some interesting results (summarized in table 1) in terms of rejection rates. The two following parts introduce the graphene based membrane and the preparation. The next part of this work focuses on the utilization of this membrane for Hg removal.

This paper is clear, well supported by the literature and the results are interesting for academic and industrial communities

 Accordingly, I recommend the publication of this paper in Membranes after minor revision

Comments:

  • Part 6: this part lacks of information/data in terms of operations, Hg removal (and the rest of minerals?)
  • The introduction part speaks about “drinking water”. Pages 4 and 5 only deal with industrial wastewater treatment. It should be interesting to present one example of drinking water treatment by nanofiltration.

Replies:

Thank you very much for every valuable comment and suggestions have given to the author, therefore the manuscript is more perfect and complete. In general, the author changed and revised the manuscript using track changes and highlighted the part is needed to show.

  • The author has added the information and data in terms of operation in part 6 of the manuscript.
  • The author has added one of the examples drinking water treatment by nanofiltration on Pages 3 of the manuscript.

I’m not sure, that in this case, nanofiltration is an appropriate treatment (removal of Hg and heavy metals, but other minerals too).

  • In part 5 or 6, data of graphene oxide based membranes should be interesting for comparison, (hydraulic properties, mean pore radius, electric surface charge, IEP, rejection rate of uncharged solute, etc.)
  • Line 234 to 243: in the literature about nanofiltration, the exclusion mechanisms are steric, electric and dielectric Two were just discussed !

Replies:

  • The author has added one Table (i.e. Table 2) in the part 6 about the performances of GO-based membrane for HG removal according to the reviewer’s suggestion.
  • The author has discussed the steric, electric, and dielectric factors on nanofiltration membrane in Line 234-243 of the manuscript.

Minor comments:

Line 27, different font sizes.

Replies:

The author has revised the font size in line 27. Thank you so much.

Reviewer 2 Report

Megawati Zunita reported a Mini Review about GO membrane for Hg removal in wastewater, which can be published after major revisions:

1, “Graphene derivative usually used to separate ion metal is graphene oxide (GO)……”. This sentence is inaccurate. Monolayer graphene with nanopores can also be used for separation of metal ions. For example: “Analytical chemistry, 2016, 88(20): 10002-10010. Physical Chemistry Chemical Physics, 2019, 21(11): 6126-6132.” Therefore, it is necessary to add some references about the separation of monolayer graphene to metal ions into this paper.

2, The toxicity of mercury is important content in this work, but this part of the content in section 2”Toxicity of mercury and its removal” is the same as before in “introduction”. So I suggest that the author refine this part of the content.

3, Membrane separation is an emerging technology because of its excellent performance, but the separation process and mechanism are complex, “In a membrane-based separation process, the membrane itself acts as a contactor which layer was passed through by the ion complexes and particle via diffusion”, this sentence is inaccurate. So I suggest author add some content into your paper for simply clarifying this mechanism. Some ref. needs to be added, for example: iScience, 2020, 24(1): 101920. Advanced Functional Materials, 2018, 28(43): 1805026. Analytical Chemistry, 2020, 92(20): 13630-13633.

4, Authors are required to carefully modify the format and language of the manuscript: for example: Confusing subscripts” Hg0, V2O5, Cl2……”, Confusing font size.

Author Response

Reviewer #2

Comments and Suggestions for Authors

Megawati Zunita reported a Mini Review about GO membrane for Hg removal in wastewater, which can be published after major revisions:

1, “Graphene derivative usually used to separate ion metal is graphene oxide (GO)……”. This sentence is inaccurate. Monolayer graphene with nanopores can also be used for separation of metal ions. For example: “Analytical chemistry, 2016, 88(20): 10002-10010. Physical Chemistry Chemical Physics, 2019, 21(11): 6126-6132.” Therefore, it is necessary to add some references about the separation of monolayer graphene to metal ions into this paper.

2, The toxicity of mercury is important content in this work, but this part of the content in section 2”Toxicity of mercury and its removal” is the same as before in “introduction”. So I suggest that the author refine this part of the content.

3, Membrane separation is an emerging technology because of its excellent performance, but the separation process and mechanism are complex, “In a membrane-based separation process, the membrane itself acts as a contactor which layer was passed through by the ion complexes and particle via diffusion”, this sentence is inaccurate. So I suggest author add some content into your paper for simply clarifying this mechanism. Some ref. needs to be added, for example: iScience, 2020, 24(1): 101920. Advanced Functional Materials, 2018, 28(43): 1805026. (Li, 2018 #150)

4, Authors are required to carefully modify the format and language of the manuscript: for example: Confusing subscripts” Hg0, V2O5, Cl2……”, Confusing font size.

Replies:

Thank you very much for every valuable comment and suggestions have given to the author, therefore the manuscript is more perfect and complete. In general, the author changed and revised the manuscript using track changes and highlighted the part is needed to show.

  1. The author has added the references of “Analytical chemistry, 2016, 88(20): 10002-10010. Physical Chemistry Chemical Physics, 2019, 21(11): 6126-6132 about the separation of monolayer graphene to metal ions in the manuscript according to the reviewer’s suggestion.
  2. The author has refined the content part 2 in the manuscript and rearranged some parts in part 1 into part 2.
  3. The author has added the references of iScience, 2020, 24(1): 101920. Advanced Functional Materials, 2018, 28(43): 1805026. (Li, 2018 #150) for simply clarifying the mechanism in the manuscript.
  4. The author has modified and checked more the format and language in the manuscript.

Round 2

Reviewer 2 Report

Acccept